# A Cancer-Specific Anti-Podoplanin Monoclonal Antibody, PMab-117-mG_2a_ Exerts Antitumor Activities in Human Tumor Xenograft Models

**DOI:** 10.3390/cells13221833

**Published:** 2024-11-06

**Authors:** Tomohiro Tanaka, Hiroyuki Suzuki, Tomokazu Ohishi, Mika K. Kaneko, Yukinari Kato

**Affiliations:** 1Department of Antibody Drug Development, Tohoku University Graduate School of Medicine, 2-1 Seiryo-machi, Aoba-ku, Sendai 980-8575, Miyagi, Japan; tomohiro.tanaka.b5@tohoku.ac.jp (T.T.); mika.kaneko.d4@tohoku.ac.jp (M.K.K.); 2Institute of Microbial Chemistry (BIKAKEN), Numazu, Microbial Chemistry Research Foundation, 18-24 Miyamoto, Numazu-shi 410-0301, Shizuoka, Japan; ohishit@bikaken.or.jp; 3Institute of Microbial Chemistry (BIKAKEN), Laboratory of Oncology, Microbial Chemistry Research Foundation, 3-14-23 Kamiosaki, Shinagawa-ku, Tokyo 141-0021, Japan

**Keywords:** cancer-specific monoclonal antibody, podoplanin, ADCC, mouse xenograft model

## Abstract

Podoplanin (PDPN) overexpression is associated with poor clinical outcomes in various tumors. PDPN is involved in malignant tumor progression by promoting invasiveness and metastasis. Therefore, PDPN is considered a promising target of monoclonal antibody (mAb)-based therapy. Because PDPN also plays an essential role in normal cells such as kidney podocytes, cancer specificity is required to reduce adverse effects on normal cells. We developed a cancer-specific mAb (CasMab) against PDPN, PMab-117 (rat IgM, kappa), by immunizing rats with PDPN-overexpressed glioblastoma cells. The recombinant mouse IgG_2a_-type PMab-117 (PMab-117-mG_2a_) reacted with the PDPN-positive tumor PC-10 and LN319 cells but not with PDPN-knockout LN319 cells in flow cytometry. PMab-117-mG_2a_ did not react with normal kidney podocytes and normal epithelial cells from the lung bronchus, mammary gland, and corneal. In contrast, one of the non-CasMabs against PDPN, NZ-1, showed high reactivity to PDPN in both tumor and normal cells. Moreover, PMab-117-mG_2a_ exerted antibody-dependent cellular cytotoxicity in the presence of effector splenocytes. In the human tumor xenograft models, PMab-117-mG_2a_ exhibited potent antitumor effects. These results indicated that PMab-117-mG_2a_ could be applied to antibody-based therapy against PDPN-expressing human tumors while reducing the adverse effects.

## 1. Introduction

Since the first monoclonal antibody (mAb) was approved by the U.S. Food and Drug Administration (FDA) in 1986, a variety of therapeutic antibodies and their derivatives have been developed together with advances in antibody engineering [1,2]. In mAb therapy for solid tumors, trastuzumab and pertuzumab were approved by the FDA for human epidermal growth factor receptor 2 (HER2)-overexpressed breast cancer in 1998 and 2012, respectively [3]. An anti-epidermal growth factor receptor (EGFR) mAb, cetuximab, was approved by the FDA for metastatic colorectal cancer in 2004 and head and neck squamous cell carcinomas (HNSCCs) in 2006 [4]. These mAbs exerted antibody-dependent cellular cytotoxicity (ADCC) and have been used in monotherapy or combination therapy with chemotherapy [5].

Although the number of naked mAb targets for solid tumors has not increased, antibody-drug conjugates (ADCs) are one of the fastest-growing formats of mAb-based solid tumor therapy [6]. ADCs possess covalently bound cytotoxic agents (payloads) via synthetic linkers, which exhibit high stability, selectivity, and favor pharmacokinetics [7]. In solid tumor therapy, trastuzumab emtansine (T-DM1) was first approved by the FDA in 2013 [8]. Since 2013, more than 60 ADCs entered clinical trials for a wide range of tumors. However, toxicity remains an essential problem in the development [6]. On-target, off-tumor toxicity is a cause of adverse effects when the target antigen is expressed in normal cells. Therefore, a better understanding and management of the tumor specificity of mAbs will be essential for further optimization.

Podoplanin (PDPN)/T1α/gp36/Aggrus is a type I transmembrane glycoprotein that contains three platelet aggregation-stimulating domains called PLAG1, PLAG2, and PLAG3 [9,10,11]. Some PLAG-like domains (PLDs) also exist, one of which is called the PLAG4 [9]. The PLAG3 and PLAG4 are modified with *O*-glycosylation, which is essential to bind to C-type lectin-like receptor 2 (CLEC-2) and PDPN-induced platelet aggregation [12,13]. The PDPN-induced platelet aggregation plays critical roles in tumor cell survival in circulation and hematogenous metastasis through the evasion from antitumor immunity [14] and promotion of embolization [15,16].

PDPN is predominantly localized in actin-rich microvilli and plasma membrane projections, such as filopodia, lamellipodia, and ruffles, where it co-localizes with ezrin, radixin, and moesin (ERM) family proteins [17,18]. The intracellular domain of PDPN contains juxtamembrane basic residues, which serve as binding sites for ERM proteins [18]. Once bound, the ezrin family proteins regulate Rho GTPase activity, facilitating actin cytoskeleton reorganization, thereby promoting cell migration, invasion, and stemness [19,20]. Interaction with ERM proteins is crucial for PDPN-induced epithelial-to-mesenchymal transition (EMT) in tumor progression as well as lymphangiogenesis and immune responses [21,22]. Additionally, two serine residues in the intracellular domain are phosphorylated by protein kinase A and cyclin-dependent kinase 5, which inhibits cell motility [23]. These findings imply that phosphorylation of the intracellular domain may influence the interaction between PDPN and ERM proteins, as well as the activation of Rho GTPase.

PDPN promotes tumor metastasis through the recruitment of the ERM complex, which remodels actin cytoskeletons and EMT [24]. The depletion of PDPN potently suppressed transforming growth factor-β (TGF-β)-induced EMT [25], indicating the critical roles of PDPN in EMT and malignant progression of tumors. Moreover, PDPN-positive tumor cells exhibit a diverse pattern of invasion, such as ameboid invasion in melanoma [26] and collective invasion in SCCs [17]. Furthermore, PDPN binds to hyaluronan receptor CD44 [27] and matrix metalloproteinases [28]. The complexes mediate the formation of tumor invadopodia, which promotes extracellular matrix (ECM) degradation and invasiveness [29]. In the clinic, high PDPN expression was associated with shortened overall survival in patients with various tumors, including HNSCCs, esophageal SCCs, gastric adenocarcinomas, gliomas, and mesotheliomas [30,31,32,33].

The elevated expression of PDPN is also observed in cancer-associated fibroblasts (CAFs), a principal constituent of the tumor microenvironment (TME) [34]. Increased abundance of PDPN in CAFs is correlated with poor clinical outcomes in pancreatic [35], breast [36], and lung [37,38,39] cancer patients. The PDPN-positive CAFs from lung tumors were reported to affect the therapeutic outcomes of EGFR inhibitors [40]. The PDPN-positive CAFs are also involved in the formation of an immunosuppressive TME through the secretion of TGF-β, which reduces antitumor immune responses [41]. Additionally, PDPN-positive CAFs were associated with low interleukin-2 activity and trastuzumab resistance in patients with HER2-positive breast cancer [42]. Therefore, PDPN in tumors and CAFs has been recognized as a useful diagnostic marker and an attractive target for tumor therapy. Since PDPN plays an essential role in normal cells such as kidney podocytes, lymphatic endothelial cells, and lung alveolar epithelial type I cells [9], anti-PDPN mAbs that recognize tumor cell-expressed PDPN but not normal cell-expressed PDPN have been desired for tumor therapy.

Our group has developed cancer-specific mAbs (CasMabs) against PDPN, which were obtained by immunization of mice with PDPN-overexpressed glioblastoma LN229 cells. LpMab-2 [43] and LpMab-23 [44] were selected by the cancer-specific reactivity in flow cytometry and immunohistochemistry. Furthermore, they were converted and produced mouse IgG_2a_ type mAbs that showed the potent ADCC and antitumor effect in xenograft models of human tumors [45,46]. In this study, we established another CasMab against PDPN (PMab-117) by immunization of a rat with PDPN-overexpressed LN229 cells. We further evaluated the ADCC activity and antitumor effect against PDPN-positive tumor cells.

## 2. Materials and Methods

### 2.1. Cell Lines and Cell Culture

LN229, HBEC3-KT, hTERT-HME1, and P3X63Ag8U.1 (P3U1) were purchased from the American Type Culture Collection (ATCC, Manassas, VA, USA). 293FT was purchased from Thermo Fisher Scientific, Inc. (Thermo; Waltham, MA, USA). PODO/TERT256 and hTCEpi were purchased from EVERCYTE (Vienna, Austria). Human glioblastoma LN319 cells were purchased from Addexbio Technologies (San Diego, CA, USA). Human lung squamous cell carcinoma PC-10 cells were purchased from Immuno-Biological Laboratories Co., Ltd. (Gunma, Japan).

PDPN-overexpressed LN229 (LN229/PDPN) cells were established as previously described [43]. LN229, LN229/PDPN, and LN319 cells were cultured in Dulbecco’s Modified Eagle’s Medium (DMEM) [Nacalai Tesque, Inc. (Nacalai), Kyoto, Japan]. PC-10 cells were cultured in Roswell Park Memorial Institute (RPMI)-1640 medium (Nacalai). These media were supplemented with 10% heat-inactivated fetal bovine serum (FBS; Thermo), 0.25 μg/mL amphotericin B, 100 μg/mL streptomycin, and 100 units/mL penicillin (Nacalai). ExpiCHO-S and Fut8-deficient ExpiCHO-S (BINDS-09) cells were cultured as described previously [45].

Immortalized normal epithelial cell lines were maintained as follows: PODO/TERT256, MCDB131 (Pan Biotech, Bayern, Germany) supplemented with GlutaMAX^TM^-I (Thermo), Bovine Brain Extract (9.6 μg/mL, Lonza, Basel, Switzerland), EGF [8 ng/mL, Sigma-Aldrich Corp. (Sigma), St. Louis, MO, USA], Hydrocortisone (20 ng/mL, Sigma), 20% FBS (Sigma), and G418 (100 μg/mL, InvivoGen, San Diego, CA, USA); HBEC3-KT, Airway Epithelial Cell Basal Medium and Bronchial Epithelial Cell Growth Kit (ATCC); hTERT-HME1, Mammary Epithelial Cell Basal Medium BulletKit^TM^ without GA-1000 (Lonza); hTCEpi, KGMTM-2 BulletKit^TM^ (Lonza).

All cell lines were cultured at 37 °C in a humidified atmosphere with 5% CO_2_ and 95% air.

### 2.2. Animals

The animal experiments aimed at establishing anti-PDPN mAbs were approved by the Animal Care and Use Committee of Tohoku University (approval no. 2016MdA-153). To evaluate the ADCC and antitumor efficacy of PMab-117-mG_2a_, animal experiments were authorized by the Institutional Committee for Experiments at the Institute of Microbial Chemistry (approval nos. 2024-062 [ADCC] and 2018-031 [antitumor activity]). Animals were housed under pathogen-free conditions with an 11-h light/13-h dark cycle, and food and water were provided ad libitum. Health and body weight were monitored every one to five days. Humane endpoints were defined as body weight loss exceeding 25% and a maximum tumor size of over 3000 mm^3^.

### 2.3. Hybridoma Production

A five-week-old Sprague-Dawley rat (CLEA Japan, Tokyo, Japan) was immunized via intraperitoneal injection with LN229/PDPN (1 × 10^9^ cells) in combination with Imject Alum (Thermo). Following three weekly injections (1 × 10^9^ cells per rat), a final booster injection (1 × 10^9^ cells per rat) was administered two days prior to spleen cell collection. Hybridomas were generated as previously described [44], and culture supernatants were screened using an enzyme-linked immunosorbent assay (ELISA) for binding to the PDPN ectodomain (PDPNec). The higher reactivity to cancer cell lines (PC-10 and LN319) than embryonic kidney 293FT cells using flow cytometry was critical for selecting CasMabs.

### 2.4. ELISA

PDPNec was coated onto Nunc Maxisorp 96-well immunoplates (Thermo) at a concentration of 1 μg/mL for 30 min. After blocking with 1% bovine serum albumin (BSA) in 0.05% Tween 20-phosphate-buffered saline (PBS, Nacalai), the plates were incubated with culture supernatant, followed by the addition of peroxidase-conjugated anti-rat immunoglobulins (Sigma) diluted 1:20,000. The enzymatic reaction was developed using the ELISA POD Substrate TMB Kit (Nacalai). Optical density was read at 655 nm using an iMark microplate reader (Bio-Rad Laboratories, Inc., Berkeley, CA, USA).

### 2.5. Antibodies

The V_H_ cDNA of PMab-117 and the C_H_ of mouse IgG_2a_ were cloned into the pCAG-Neo vector [FUJIFILM Wako Pure Chemical Corporation (Wako), Osaka, Japan]. Similarly, V_L_ cDNA of PMab-117 and the C_L_ of the mouse kappa chain were cloned into the pCAG-Ble vector (Wako). We introduced the PMab-117-mG_2a_ expression vectors into BINDS-09 cells using the ExpiCHO-S Expression System (Thermo). We purified PMab-117-mG_2a_ using Ab-Capcher (ProteNova Co., Ltd., Kagawa, Japan). NZ-1 (an anti-PDPN mAb) [47] and PMab-231 (control mouse IgG_2a_) [48] were previously described. Mouse IgG (mIgG) was purchased from Wako.

### 2.6. Flow Cytometry

Cells were harvested using 0.25% trypsin and 1 mM ethylenediamine tetraacetic acid (EDTA; Nacalai). The cells (1 × 10^5^ cells/sample) were incubated with NZ-1, PMab-117, PMab-117-mG_2a_, or blocking buffer (control, 0.1% BSA in PBS) for 30 min at 4 °C. Subsequently, the cells were incubated with Alexa Fluor 488-conjugated anti-rat or mouse IgG (1:1000; Cell Signaling Technology, Danvers, MA, USA) for 30 min at 4 °C. Fluorescence data were acquired using the SA3800 Cell Analyzer (Sony Corp., Tokyo, Japan) and analyzed with FlowJo software (BD Biosciences, Franklin Lakes, NJ, USA).

### 2.7. Determination of the Binding Affinity by Flow Cytometry

After being suspended in 100 μL of serially diluted PMab-117-mG_2a_ or NZ-1, the cells were then incubated with Alexa Fluor 488-conjugated anti-mouse or rat IgG (1:200), respectively. The SA3800 Cell Analyzer was used to obtain the fluorescence data. To calculate the dissociation constant (*K*_D_), GraphPad PRISM 6 software (GraphPad Software, Inc., La Jolla, CA, USA) was used.

### 2.8. ADCC

Effector splenocytes were obtained from the spleen of female BALB/c nude mice (Jackson Laboratory Japan, Inc., Kanagawa, Japan). LN229/PDPN, PC-10, and LN319 cells were labeled with 10 µg/mL of Calcein AM (Thermo). Target cells (1 × 10^4^ cells/well) were plated and mixed with the effector cells (effector-to-target ratio, 50:1) and 100 μg/mL of control mouse IgG_2a_ (PMab-231) or PMab-117-mG_2a_. The calcein released into the medium was measured following a 4.5 h incubation. Fluorescence intensity was assessed using a microplate reader (Power Scan HT; BioTek Instruments, Winooski, VT, USA) with excitation and emission wavelengths set to 485 nm and 538 nm, respectively. After lysing all cells with a buffer containing 0.5% Triton X-100, 10 mM Tris-HCl (pH 7.4), and 10 mM EDTA, cytotoxicity (% lysis) was determined using the formula % lysis = (E − S)/(M − S) × 100, where E represents the fluorescence of both target and effector cells, S is the spontaneous fluorescence of the target cells alone, and M is the maximum fluorescence observed.

### 2.9. Antitumor Activity of PMab-117-mG_2a_ in Xenografts of LN229/PDPN, PC-10, and LN319

LN229/PDPN, PC-10, or LN319 was suspended in 0.3 mL of DMEM (1.33 × 10^8^ cells/mL) and mixed with 0.5 mL of BD Matrigel Matrix Growth Factor Reduced (BD Biosciences). Then, BALB/c nude mice (Jackson Laboratory Japan, Kanagawa, Japan) were injected subcutaneously in the left flank with 100 μL of the suspension (5 × 10^6^ cells). Following the inoculation of LN229/PDPN, PC-10, or LN319 (day 0), PMab-117-mG_2a_ (n = 8) or control mIgG (n = 8) was intraperitoneally injected into the xenograft-bearing mice on days 1, 8, and 16 (LN229/PDPN and LN319) or days 1, 8, 14, and 22 (PC-10). The tumor volume was calculated using the following formula: volume = W^2^ × L/2, where W is the short diameter and L is the long diameter. All mice were euthanized by cervical dislocation 22~30 days after cell inoculation.

### 2.10. Statistical Analyses

Data are presented as the mean ± standard error of the mean (SEM). A two-tailed unpaired *t*-test was employed for statistical analysis of ADCC and tumor weight, while two-way ANOVA followed by Sidak’s multiple comparisons test was used for tumor volume and mouse weight. A *p*-value of less than 0.05 was considered statistically significant. We used GraphPad PRISM 6 software for the calculation.

## 3. Results

### 3.1. Production and Screening of an Anti-PDPN CasMab, PMab-117

We immunized a rat with LN229/PDPN cells. The culture supernatants of hybridomas were screened using ELISA with PDPNec. We further screened the reactivity to PDPN-positive cancer cell lines (PC-10 and LN319) and embryonic kidney 293FT cells using flow cytometry (Figure 1A). One of the established hybridomas, PMab-117 (IgM, kappa) reacted with LN229/PDPN, PC-10, and LN319, but not with PDPN-negative LN229 and PDPN-knockout LN319 (BINDS-55) (Figure 1B). NZ-1, an anti-PDPN mAb (rat IgG_2a_), showed a higher reactivity to those cancer cell lines (Figure 1B). Next, we compared the reactivity of PMab-117 and NZ-1 to 293FT and PODO/TERT256 (TERT-expressed normal kidney podocyte). As shown in Figure 1C, PMab-117 exhibited a low and no reactivity to 293FT and PODO/TERT256, respectively. In contrast, NZ-1 showed the reactivity to both 293FT and PODO/TERT256 (Figure 1C).

### 3.2. Production of PMab-117-mG_2a_ and the Reactivity to Cancer Cells, Normal Kidney Podocytes, and Epithelial Cells

Since PMab-117 is an IgM mAb, comparing the reactivity to IgG mAbs, including NZ-1, is somewhat problematic. Furthermore, evaluating in vivo antitumor activity in mouse xenograft models is difficult. Therefore, we produced a class-switched mouse IgG_2a_ mAb (PMab-117-mG_2a_) from PMab-117. We cloned the V_H_ cDNA of PMab-117 and combined it with the C_H_ cDNA of mouse IgG_2a_. We also cloned the V_L_ cDNA of PMab-117 and combined it with the C_L_ cDNA of the mouse kappa light chain. Finally, PMab-117-mG_2a_ was produced using Fut8-deficient ExpiCHO-S (BINDS-09) cells (Figure 2A). In reduced conditions, we confirm the purity of original and recombinant mAbs by SDS-PAGE (Appendix A). As shown in Figure 2B, PMab-117-mG_2a_ reacted with LN229/PDPN, PC-10, and LN319, but not with LN229 and BINDS-55. NZ-1 showed a similar reactivity to those cancer cell lines (Figure 2B). We next compared the reactivity of PMab-117-mG_2a_ and NZ-1 to 293FT, PODO/TERT256, and TERT-expressed normal epithelial cells, including HBEC3-KT (lung bronchus), hTERT-HME1 (mammary gland), and hTCEpi (cornea). As shown in Figure 2C, PMab-117-mG_2a_ exhibited a low reactivity to 293FT. Furthermore, PMab-117-mG_2a_ did not show reactivity to PODO/TERT256, HBEC3-KT, hTERT-HME1, and hTCEpi. In contrast, NZ-1 showed reactivity to 293FT and those normal cells (Figure 2C).

The *K*_D_ for the interaction of PMab-117-mG_2a_ and NZ-1 with LN319 was determined by flow cytometry. The *K*_D_ values for PMab-117-mG_2a_ and NZ-1 with LN319 were 1.9 × 10^−7^ M (Figure 3A) and 5.0 × 10^−9^ M (Figure 3B), respectively.

These results indicated that PMab-117-mG_2a_ could recognize tumor cells but not normal kidney podocytes and epithelial cells from the lung bronchus, mammary gland, and cornea. In contrast, one of the non-CasMabs against PDPN, NZ-1, showed high reactivity to both tumor and normal epithelial cells.

### 3.3. ADCC by PMab-117-mG_2a_ Against PDPN-Positive Cells

We then examined whether PMab-117-mG_2a_ possesses ADCC activity against PDPN-positive cells. As shown in Figure 4, PMab-117-mG_2a_ induced ADCC against LN229/PDPN cells (17.3% cytotoxicity; *p* < 0.01) more effectively than the control mouse IgG_2a_ (3.8% cytotoxicity). PMab-117-mG_2a_ also elicited more potent ADCC against endogenous PDPN expressing tumor PC-10 (42.1% cytotoxicity; *p* < 0.01) and LN319 (23.9% cytotoxicity; *p* < 0.01) cells. These results demonstrated that PMab-117-mG_2a_ exhibited potent ADCC activities against PDPN-positive cells.

### 3.4. Antitumor Effects of PMab-117-mG_2a_ Against PDPN-Positive Cells in Mouse Xenograft Models

Following the inoculation of LN229/PDPN, PC-10, or LN319 (day 0), PMab-117-mG_2a_ or control mIgG was intraperitoneally injected into the xenograft-bearing mice on days 1, 8, and 16 (LN229/PDPN and LN319) or days 1, 8, 14, and 22 (PC-10). The tumor volume was measured on the indicated days. The PMab-117-mG_2a_ administration resulted in a significant reduction in LN229/PDPN xenografts on days 16 (*p* < 0.01), 27 (*p* < 0.01), and 30 (*p* < 0.01) compared with that of control mIgG (Figure 5A). A significant reduction was observed in the PC-10 xenograft on days 22 (*p* < 0.01), 26 (*p* < 0.01), and 28 (*p* < 0.01) (Figure 5B). A significant reduction was also observed in the LN319 xenograft on days 19 (*p* < 0.01) and 22 (*p* < 0.01) (Figure 5C).

A significant reduction in xenograft weight caused by PMab-117-mG_2a_ was observed in LN229/PDPN (64% reduction; *p* < 0.01; Figure 5D), PC-10 (55% reduction; *p* < 0.01; Figure 5E), and LN319 (48% reduction; *p* < 0.01; Figure 5F). The LN229/PDPN, PC-10, and LN319 xenografts were resected from mice on days 30, 28, and 22, respectively (Figure 5G–I).

The xenograft-bearing mice did not lose body weight (Figure 5J–L). The mice on day 30 (LN229/PDPN), day 28 (PC-10), and day 22 (LN319) are shown in Appendix A.

## 4. Discussion

For the development of mAbs for tumor therapy, the identification and validation of adequate antigenic targets are important [1]. To achieve an acceptable therapeutic index and avoid on-target toxicity, target antigens should ideally have high tumor expression levels and little or no expression in normal tissues. However, the limitation of the ideal target antigens is a severe problem. Some technologies, including bispecific antibodies, defucosylated antibodies, and ADCs, enhance the activity of antibodies, contributing to tumor therapy development. However, on-target toxicity due to recognizing antigens in normal cells has not been resolved. Therefore, selecting mAb that recognizes cancer-specific epitopes is essential to reduce the adverse effects. This study developed a novel CasMab against PDPN (PMab-117) by immunizing LN229/PDPN with a rat (Figure 1). The mouse IgG_2a_ type PMab-117 (PMab-117-mG_2a_) reacted with the PDPN-positive tumor cells but not with normal kidney podocytes and normal epithelial cells from lung bronchus, mammary gland, and corneal (Figure 2). Furthermore, PMab-117-mG_2a_ exerted a potent ADCC (Figure 4) and antitumor effect in PC-10 and LN319 xenografts (Figure 5).

The reactivity of PMab-117-mG_2a_ in flow cytometry is low in PC-10 (Figure 2B). In contrast, PMab-117-mG_2a_ exhibited high ADCC activity (Figure 4) and antitumor effect (Figure 5). Although the target cell-derived immunosuppressive factors such as PD-L1 or TGF-β would contribute to the responses, the reactivity of PMab-117-mG_2a_ in PC-10 is sufficient to exert ADCC and antitumor efficacy in vivo. In this condition, PMab-117-mG_2a_ did not react with normal kidney podocytes, lung bronchus epithelial cells, mammary gland epithelial cells, and corneal epithelial cells (Figure 2C). We should investigate the in vivo side effects in the future. Human PDPN PLAG4 domain knock-in mice were generated [49]. Since PMab-117-mG_2a_ possesses the epitope around the PLAG4 domain (see below), it is worthwhile to evaluate the side effect in vivo if PMab-117-mG_2a_ can recognize the human/mouse chimeric PDPN. Furthermore, evaluating the humanized PMab-117 to apply clinical studies is essential. We should assess not only the antitumor efficacy but also toxicity to normal tissues using cynomolgus monkeys [44].

We have reported CasMabs against PDPN (LpMab-2 [43] and LpMab-23 [44]), which were obtained by immunization of LN229/PDPN with mice. LpMab-2 recognizes a glycopeptide (Thr55-Leu64) structure of PDPN [43]. LpMab-23 recognizes a naked peptide structure of PDPN (Gly54–Leu64), especially Gly54, Thr55, Ser56, Glu57, Asp58, Arg59, Tyr60, and Leu64 of PDPN is a critical epitope of LpMab-23 [50]. PMab-117, obtained by immunization of LN229/PDPN with a rat, recognizes the glycopeptide structure of PDPN (Ile78-Thr85) around PLAG4 domain, which includes *O*-glycosylated Thr85 [9]. A mAb with the specificity and epitope of PMab-117 has never been obtained by immunization with mice. Therefore, the strategies for CasMab generation using mouse or rat immunization can contribute to developing novel CasMabs against various tumor antigens.

Chimeric antigen receptor (CAR)-T cell therapy targeting solid tumors has been assessed in clinical trials [51]. Our anti-PDPN mAbs, NZ-1 and LpMab-2, were developed specifically for CAR-T cell therapy and evaluated in preclinical studies. Systemic administration of NZ-1-based CAR-T cells inhibited intracranial glioma growth in immunodeficient mice [52]. Similarly, LpMab-2-based CAR-T cells killed patient-derived glioma stem cells and suppressed the growth of a glioma xenograft in immunodeficient mice [53]. Consequently, CAR-T cell therapy targeting PDPN shows promise as a potential immunotherapy for treating glioblastoma [54]. It is essential to explore the cancer-specific reactivity of the PMab-117 single-chain Fv and apply it to CAR-T cell therapy.

Our developed CasMabs against HER2 (clones H_2_Mab-214 [55] and H_2_Mab-250 [56]) were also screened by the reactivity to cancer and normal cells in flow cytometry. Both CasMabs exhibited the antitumor effect in mouse xenograft models using their recombinant mouse IgG_2a_ or human IgG_1_ mAbs [48,57]. H_2_Mab-250 has been developed as CAR-T-cell therapy. The phase I study has been conducted in the US (NCT06241456). Furthermore, the recognition mode of H_2_Mab-214 was solved by X-ray crystallography [55]. H_2_Mab-214 recognizes a locally misfolded structure of HER2 extracellular domain 4, which usually forms a β-sheet [55]. The structural analysis of the PMab-117-PDPN complex is also essential to reveal the mechanism of cancer-specific recognition.

As shown in Figure 3, PMab-117-mG_2a_ possesses ~40-fold lower affinity (*K*_D_: 1.9 × 10^−7^ M) than NZ-1 (*K*_D_: 5.0 × 10^−9^ M). The *K*_D_ values of other CasMabs against PDPN (LpMab-2 and LpMab-23) were previously determined as 5.7 × 10^−9^ M and 1.2 × 10^−8^ M, respectively [43,50]. These CasMabs have different binding affinities ranging from 10^−7^ M to 10^−9^ M. Recently, CAR’s affinity for the antigen determines CAR-T therapy’s efficacy and persistence. Trogocytosis was first proposed as a mechanism of immune escape of CAR-T therapy against CD19 [58]. When CD19-positive lymphoma cells are co-cultured with CAR-T cells equipped with the high-affinity anti-CD19 FMC63-based CAR, the CAR-T cells remove CD19 from lymphoma cells and incorporate it into their plasma membrane [58]. This process, known as “trogocytosis”, results in the generation of antigen-negative target cells. Additionally, CAR-T cells that acquire CD19 through trogocytosis can be targeted by other CAR-T cells [58]. To mitigate trogocytosis, reducing CAR affinity has been suggested. In two clinical trials, CD19-targeting CARs with approximately 40-fold lower affinity than the FMC63-based CAR demonstrated greater efficacy and persistence compared to FMC63-based CAR-T cells [59,60]. These findings indicate that lowering CAR affinity can reduce trogocytosis while preserving antitumor activity and clinical effectiveness. The characteristics of anti-PDPN CasMabs may aid in the future design of PDPN-targeting CAR-T cells by minimizing trogocytosis and maintaining cancer specificity.

ADCs are one of the growing modalities for solid tumor therapy. However, safety issues are one of the reasons for their termination. Bivatuzumab–mertansine, a humanized anti-CD44v6 ADC, was developed and evaluated in clinical trials [61]. However, the clinical trials were terminated due to its severe skin toxicity. Since CD44v6 is expressed in the skin epidermis, the efficient accumulation of mertansine in the skin probably leads to skin disorders [61,62]. The strategy of CasMab selection may contribute to developing anti-CD44v6 CasMabs to reduce the adverse effects and overcome the depletion of target antigens for tumor therapy.

## 5. Conclusions

A novel CasMab against PDPN, PMab-117-mG_2a_ possesses a superior reactivity to cancer cells, but not to normal cells. PMab-117-mG_2a_ exerted potent antitumor effects in human xenograft model, and is expected to be applicable to human cancer treatment by generating a humanized mAb, ADC, and CAR-T.

## Figures and Tables

**Figure 1 cells-13-01833-f001:**
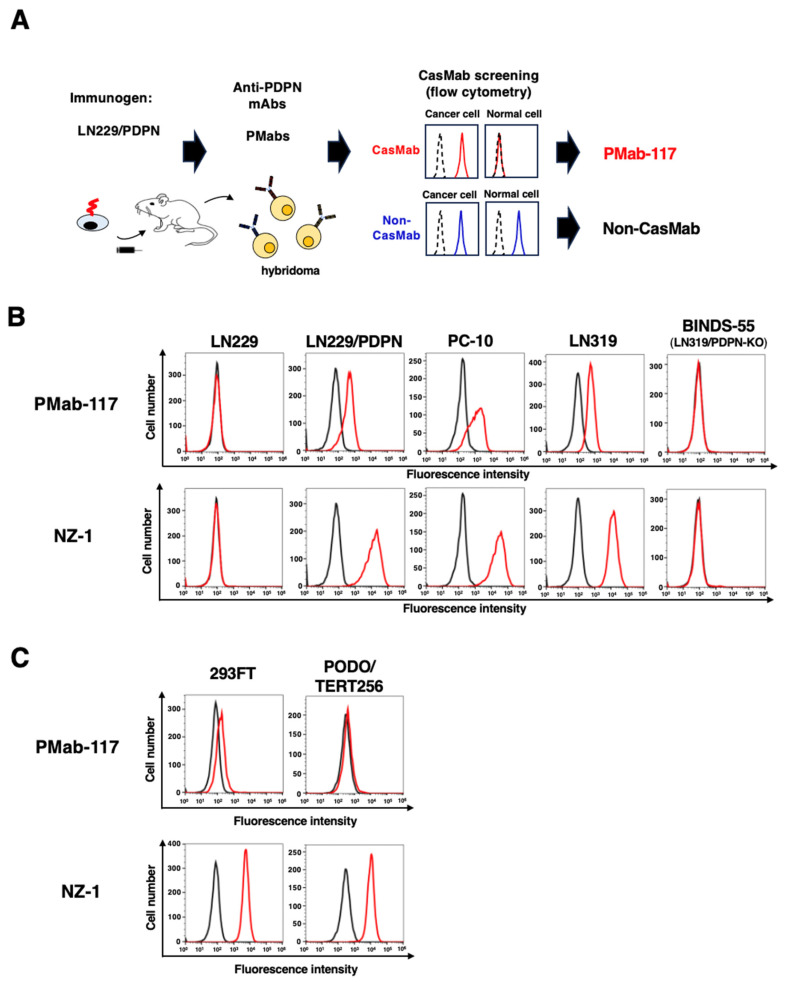
Selection of anti-PDPN CasMab, PMab-117. (**A**) A scheme of CasMab selection from anti-PDPN mAb-producing hybridoma clones. (**B**) Flow cytometry using PMab-117 (10 μg/mL; Red line), NZ-1 (10 μg/mL; Red line), or buffer control (Black line) against LN229, LN229/PDPN, PC-10, LN319, and PDPN-knockout LN319 (BINDS-55). (**C**) Flow cytometry using PMab-117 (10 μg/mL; Red line), NZ-1 (10 μg/mL; Red line), or buffer control (Black line) against 293FT (human embryonic kidney) and PODO/TERT256 (TERT-expressed normal kidney podocyte).

**Figure 2 cells-13-01833-f002:**
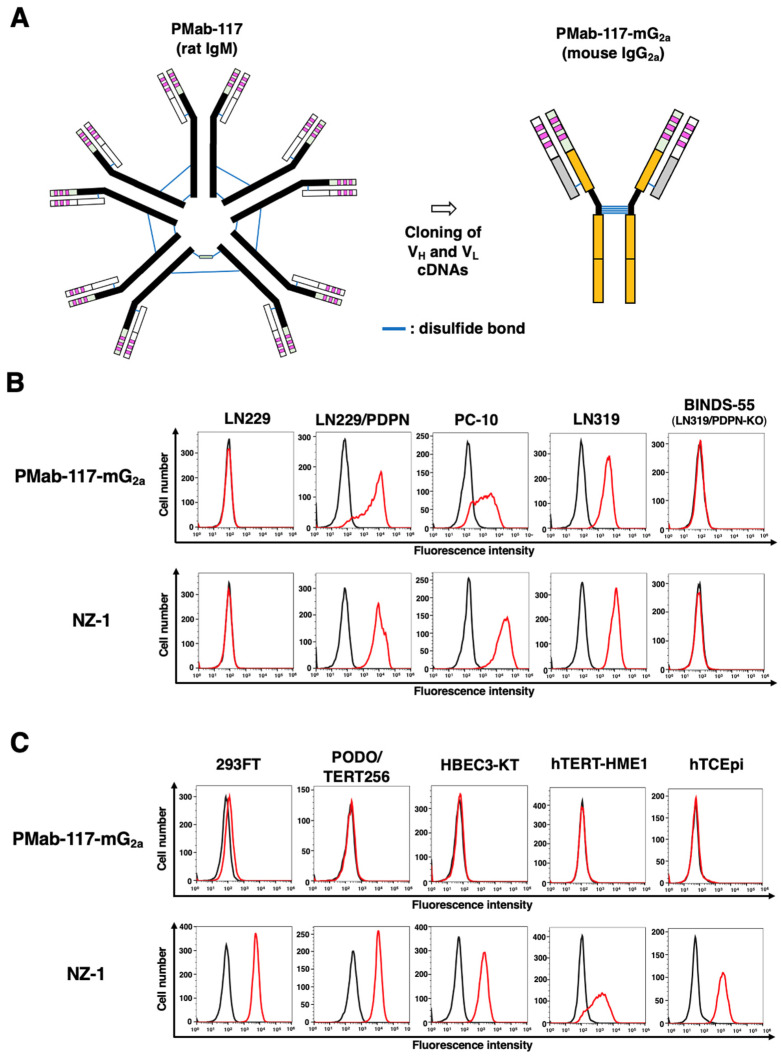
Production of PMab-117-mG_2a_ and reactivity to cancer cells, normal kidney podocytes, and epithelial cells. (**A**) Class-switched mouse IgG_2a_ mAb, PMab-117-mG_2a,_ was generated from PMab-117 (rat IgM). (**B**) Flow cytometry using PMab-117-mG_2a_ (1 μg/mL; Red line), NZ-1 (1 μg/mL; Red line), or buffer control (Black line) against LN229, LN229/PDPN, PC-10, LN319, and PDPN-knockout LN319 (BINDS-55). (**C**) Flow cytometry using PMab-117-mG_2a_ (1 μg/mL; Red line), NZ-1 (1 μg/mL; Red line) or buffer control (Black line) against 293FT (human embryonic kidney), PODO/TERT256 (kidney podocyte), HBEC3-KT (lung bronchus epithelial cell), hTERT-HME1 (mammary gland epithelial cell), and hTCEpi (corneal epithelial cell).

**Figure 3 cells-13-01833-f003:**
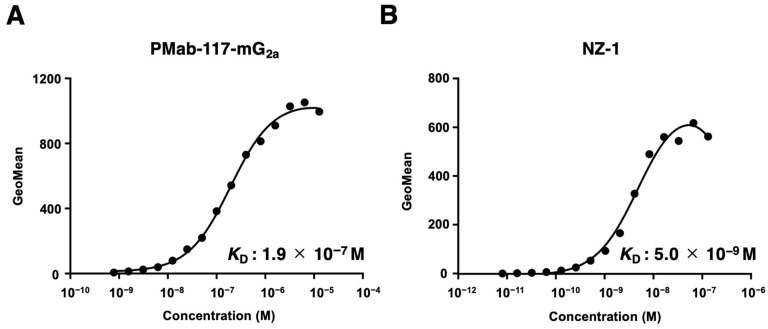
Determination of the binding affinity of PMab-117-mG_2a_ and NZ-1 using flow cytometry. LN319 cells were suspended in PMab-117-mG_2a_ (**A**) or NZ-1 (**B**) at indicated concentrations, followed by treatment with anti-mouse or rat IgG conjugated with Alexa Fluor 488. The SA3800 Cell Analyzer was used to analyze fluorescence data. The dissociation constant (*K*_D_) values were determined using GraphPad Prism 6.

**Figure 4 cells-13-01833-f004:**
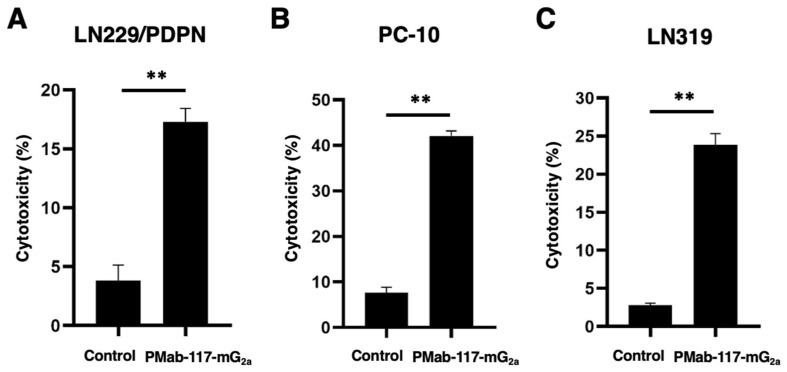
ADCC activity by PMab-117-mG_2a_ in PDPN-positive cells. ADCC induced by PMab-117-mG_2a_ or control mouse IgG_2a_ (PMab-231) against LN229/PDPN (**A**), PC-10 (**B**), and LN319 (**C**) cells. Values are shown as the mean ± SEM. Asterisks indicate statistical significance (** *p* < 0.01; two-tailed unpaired *t*-test).

**Figure 5 cells-13-01833-f005:**
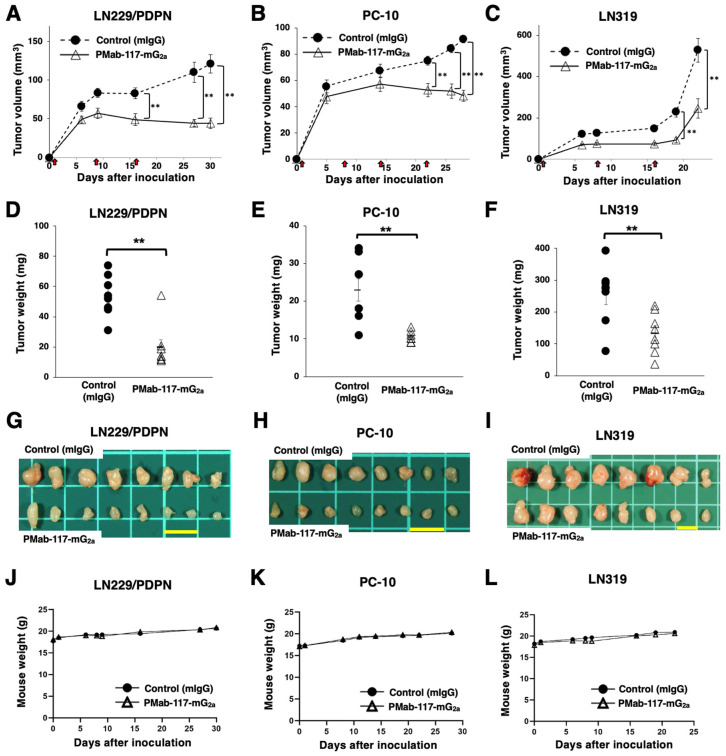
Antitumor activity of PMab-117-mG_2a_ against human tumor xenografts. (**A**–**C**) LN229/PDPN (**A**), PC-10 (**B**), and LN319 (**C**) cells were subcutaneously injected into BALB/c nude mice (day 0). PMab-117-mG_2a_ (100 μg) or control mouse IgG (mIgG, 100 μg) were intraperitoneally injected into each mouse on days 1, 8, and 16 (LN229/PDPN and LN319, arrows) or days 1, 8, 14, and 22 (PC-10, arrows). The tumor volume is represented as the mean ± SEM. ** *p* < 0.01 (two-way ANOVA and Sidak’s multiple comparisons test). (**D**–**F**) The mice were euthanized on day 30 (LN229/PDPN), day 28 (PC-10), or day 22 (LN319) after cell inoculation. The tumor weights of LN229/PDPN (**D**), PC-10 (**E**), and LN319 (**F**) xenografts were measured. Values are presented as the mean ± SEM. ** *p* < 0.01, (two-tailed unpaired *t*-test). (**G**–**I**) LN229/PDPN (**G**), PC-10 (**H**), and LN319 (**I**) xenograft tumors (scale bar, 1 cm). (**J**–**L**) Body weights of LN229/PDPN (**J**), PC-10 (**K**), and LN319 (**L**) xenograft-bearing mice treated with control mIgG or PMab-117-mG_2a_.

## Data Availability

The data presented in this study are available in the article and Appendix A.

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
