# Peer review of "A Cancer-Specific Anti-Podoplanin Monoclonal Antibody, PMab-117-mG2a Exerts Antitumor Activities in Human Tumor Xenograft Models"

_cells, 2024, doi:10.3390/cells13221833_

Round 1

Reviewer 1 Report

Comments and Suggestions for Authors

Dear authors:

Their article entitled A Cancer-Specific Anti-Podoplanin Monoclonal Antibody, 2 PMab-117-mG2a Exerted Antitumor Activity in Human Tumor 3 Xenograft Models, is extremely interesting since it proposes PDPN as a marker of a poor prognosis in different types of cancer. Considering that PDPN has great potential in ​​oncology and could even be an excellent therapeutic target, the results show that it is highly specific to tumor cells, which is an important advantage over chemotherapy and radiotherapy. Regarding your proposal, I recommend reviewing the following points please:

I recommend that you improve your summary, especially in the results and conclusions.

In the introduction: Consider that they should improve the syntax and have better coherence in the foundations they present for their work and in this way be more friendly to the reader.    I also recommend that your references be more current.   I recommend going too long in the first paragraphs commenting on the history of monoclonal antibodies since it goes out of context of the main topic. Apart from line 67, I suggest delving deeper into the biochemical and signaling mechanisms of PDPN and dividing what has been found in cell lines, in animal models and in humans.

In the materials and methods section:

Were the cell lines you mentioned new or how many approximate passages did they have?   Under what conditions were they used for the experiments?

How specific (affinity) are the antibodies used in your performed techniques?

What statistical program did you use? Please describe your statistical analysis in more detail.    Because they used standard error and not standard deviation.   Why they applied Sidak's multiple comparisons test. Did you calculate the distribution of your quantitative variables? What statistical power does your study have or how did you validate it in relation to your sample size?   Did you observe adverse effects? What would you expect in its possible application in humans?

Discussion

Please reorganize this section, first discussing your results in order of importance, contrast your observations with the most current literature, comment on your perspectives and do not mix the discussion with the conclusions, the latter must be extremely punctual.

Conclusions

Your conclusions must be rewritten, they need to be more punctual, focusing on what your general objective was and on the specific objectives.

Comments on the Quality of English Language

I detect errors in the grammar of your writing, I recommend improving this aspect and having it supervised by a specialized  person.

Author Response

Their article entitled A Cancer-Specific Anti-Podoplanin Monoclonal Antibody, PMab-117-mG2a Exerted Antitumor Activity in Human Tumor Xenograft Models, is extremely interesting since it proposes PDPN as a marker of a poor prognosis in different types of cancer. Considering that PDPN has great potential in oncology and could even be an excellent therapeutic target, the results show that it is highly specific to tumor cells, which is an important advantage over chemotherapy and radiotherapy. Regarding your proposal, I recommend reviewing the following points please:

I recommend that you improve your summary, especially in the results and conclusions.

In the introduction: Consider that they should improve the syntax and have better coherence in the foundations they present for their work and in this way be more friendly to the reader. I also recommend that your references be more current. I recommend going too long in the first paragraphs commenting on the history of monoclonal antibodies since it goes out of context of the main topic. Apart from line 67, I suggest delving deeper into the biochemical and signaling mechanisms of PDPN and dividing what has been found in cell lines, in animal models and in humans.

First, we condense the second paragraph. Furthermore, we added the new (4th) paragraph about the biochemical and signaling mechanisms of PDPN, as follows.

PDPN is predominantly localized in actin-rich microvilli and plasma membrane projections, such as filopodia, lamellipodia, and ruffles, where it co-localizes with ezrin, radixin, and moesin (ERM) family proteins [15]. The intracellular domain of PDPN contains juxtamembrane basic residues, which serve as binding sites for ERM proteins [15]. Once bound, the ezrin family proteins regulate Rho GTPase activity, facilitating actin cytoskeleton reorganization, thereby promoting cell migration, invasion, and stemness [16,17]. Interaction with ERM proteins is crucial for PDPN-induced epithelial-to-mesenchymal transition (EMT) in tumor progression as well as lymphangiogenesis and immune responses [18]. Additionally, two serine residues in the intracellular domain are phosphorylated by protein kinase A and cyclin-dependent kinase 5, which inhibits cell motility [19]. These findings imply that phosphorylation of the intracellular domain may influence the interaction between PDPN and ERM proteins, as well as the activation of Rho GTPase.

In the materials and methods section:

Were the cell lines you mentioned new or how many approximate passages did they have? Under what conditions were they used for the experiments?

PDPN-overexpressed LN229 was newly established in our laboratory.

P3X63Ag8U.1 (P3U1) was used less than 20 passages.

Cancer cell lines (LN229, LN319, and PC-10) were less than 10 passages.

Normal epithelial cell lines were used in less than 5 passages.

They are used in exponentially growing conditions.

How specific (affinity) are the antibodies used in your performed techniques?

As shown in Figures 1B and 2B, PMab-117 or PMab-117-mG2a reacted with LN229/PDPN, PC-10, and LN319 but not with parental LN229 and BINDS-55 (PDPN-knockout LN319). From these results, PMab-117 or PMab-117-mG2a specifically recognizes PDPN.

What statistical program did you use? Please describe your statistical analysis in more detail. Because they used standard error and not standard deviation.

We used GraphPad PRISM 6 software for the calculation. We added the information in 2.10. Statistical Analyses.

Why they applied Sidak's multiple comparisons test.

The GraphPad software recommended Sidak's multiple comparisons test as an optimal method for analyzing the results. In our animal study, we mainly used the test for statistical analysis in previously published papers.

Did you calculate the distribution of your quantitative variables?

We could not understand your questions here. Please clarify what you mean if you need our further corrections.

How did you validate it in relation to your sample size?

We used 8 animals in each group (Figure 5). Since many studies performed these kinds of experiments using around 5 animals, our sample size is considered enough.

Did you observe adverse effects?

No. This is probably because PMab-117-mG2a does not recognize mouse PDPN.

What would you expect in its possible application in humans?

We added the expectation in the conclusions, as follows.

A novel CasMab against PDPN, PMab-117-mG2a possesses a superior reactivity to cancer cells, but not to normal cells. PMab-117-mG2a exerted potent antitumor effects in human xenograft model, and is expected for the application human cancer treatment by generating a humanized mAb, ADC, and CAR-T.

Discussion

Please reorganize this section, first discussing your results in order of importance, contrast your observations with the most current literature, comment on your perspectives and do not mix the discussion with the conclusions, the latter must be extremely punctual.

We reorganized the section. We avoided a mixture of discussion and conclusions.

Conclusions

Your conclusions must be rewritten, they need to be more punctual, focusing on what your general objective was and on the specific objectives.

We newly added the conclusion session (below).

A novel CasMab against PDPN, PMab-117-mG2a possesses a superior reactivity to cancer cells, but not to normal cells. PMab-117-mG2a exerted potent antitumor effects in human xenograft model, and is expected for the application human cancer treatment by generating a humanized mAb, ADC, and CAR-T.

Reviewer 2 Report

Comments and Suggestions for Authors

This manuscript describes on the attempt for the preparation of anti-PDPN Ab specific to tumor cells, and showed its therapeutic potential. The article is basically well written, and the results are interesting. The reviewer has only minor comments.

Comments.

1.     They already developed some anti-PDPN Abs having anti-tumor activity and reported. Are there any difference in anti-tumor effects between newly developed Ab and previously developed mAbs?

2.     Is it possible to check the anti-human tumor cell activity of anti-PDPN Ab using the humanized PMab-117 Ab and human effector splenocyte cell line in vitro?

3.     Line 301: “respectively” is a mistype? (already we see it at line 300)

Comments on the Quality of English Language

 Line 301: “respectively” is a mistype? (already we see it at line 300)

Author Response

This manuscript describes on the attempt for the preparation of anti-PDPN Ab specific to tumor cells, and showed its therapeutic potential. The article is basically well written, and the results are interesting. The reviewer has only minor comments.

Comments.

  1. They already developed some anti-PDPN Abs having anti-tumor activity and reported. Are there any difference in anti-tumor effects between newly developed Ab and previously developed mAbs?

We have three CasMabs against PDPN (LpMab-2, LpMab-23, and PMab-117). in vivo antitumor efficacy was similar using those CasMabs. As we mentioned in the discussion, they have different epitopes.

LpMab-2 recognizes a glycopeptide structure of PDPN (Thr55-Leu64).

LpMab-23 recognizes a naked peptide structure of PDPN (Gly54–Leu64).

PMab-117 recognizes the glycopeptide structure of PDPN (Ile78-Thr85), which includes O-glycosylated Thr85.

  1. Is it possible to check the anti-human tumor cell activity of anti-PDPN Ab using the humanized PMab-117 Ab and human effector splenocyte cell line in vitro?

Yes. We plan to generate humanized PMab-117. In the future study, we will investigate the ADCC activity against human cancer cells and the antitumor effect in the presence of human NK cells. We included this in our conclusions, as follows.

A novel CasMab against PDPN, PMab-117-mG2a, possesses a superior reactivity to cancer cells, but not to normal cells. PMab-117-mG2a exerted potent antitumor effects in human xenograft model, and is expected for the application human cancer treatment by generating a humanized mAb, ADC, and CAR-T.

  1. Line 301: “respectively” is a mistype? (already we see it at line 300)

Thank you very much. We corrected.

Round 2

Reviewer 1 Report

Comments and Suggestions for Authors

Dear authors, I have reviewed your modified version and I agree with most of its changes, however, I hope you can respond to the following considerations:

Why they applied Sidak's multiple comparisons test.

GraphPad software recommended Sidak's multiple comparisons test as an optimal method for analyzing the results. In our animal study, we used tests for statistical analysis in previously published papers.

-This answer is not valid, each statistical test must be chosen according to the design of your study and the variables involved, could you please, with these foundations, explain the choice of this statistical test for your analysis of results and not some other test post hoc? Did you calculate the distribution of your quantitative variables?

We could not understand your questions here. Please clarify what you mean if you need our further corrections.

-To choose a statistical test where a qualitative variable and a quantitative variable are related, one of the assumptions of choice is the distribution of the sample (quantitative variable), which is usually obtained when performing a test in this case of Shapiro–Wilk; considering this result, the parametric or nonparametric test to be applied must be chosen. You assume that the distribution is normal because of the test you applied, but how do you support it? How did you validate it in relation to your sample size?

We used 8 animals in each group (Figure 5). Since many studies have performed these types of experiments using around 5 animals, our sample size is considered sufficient.

-This answer is not sufficient from a methodological and biostatistical point of view, especially in experimental studies with biological reagents, a determination of the validity of your study must be made by determining its statistical power which must be 80% minimum. Please determine the statistical power of the study.

Comments on the Quality of English Language

I recommend that the English writing should be reviewed by an expert.

Round 3

Reviewer 1 Report

Comments and Suggestions for Authors

The answers and improvements are enough, congratulations for your research